# Influence of Low-Molecular-Weight Esters on Melt Spinning and Structure of Poly(lactic acid) Fibers

**DOI:** 10.3390/ma17061268

**Published:** 2024-03-09

**Authors:** Karolina Gzyra-Jagieła, Konrad Sulak, Zbigniew Draczyński, Sławomir Kęska, Michał Puchalski, Longina Madej-Kiełbik

**Affiliations:** 1Lukasiewicz Research Network—Lodz Institute of Technology, 19/27 M. Sklodowskiej-Curie Str., 90-570 Lodz, Poland; konrad.sulak@lit.lukasiewicz.gov.pl (K.S.); slawomir.keska@lit.lukasiewicz.gov.pl (S.K.); longina.madej-kielbik@lit.lukasiewicz.gov.pl (L.M.-K.); 2Textile Institute, Lodz University of Technology, 116 Żeromskiego Street, 90-924 Lodz, Poland; zbigniew.draczynski@p.lodz.pl (Z.D.); michal.puchalski@p.lodz.pl (M.P.)

**Keywords:** poly(lactic acid) fibers, textile materials, triethyl citrate, bis (2-ethylhexyl) adipate, crystallization, supramolecular structure

## Abstract

Poly(lactic acid) has great potential in sectors where degradability is an important advantage due to its polymer nature. The medical, pharmaceutical, and packaging industries have shown interest in using PLA. To overcome the limitations of stiffness and brittleness in the polymer, researchers have conducted numerous modifications to develop fibers with improved properties. One such modification involves using plasticizing modifiers that can provide additional and desired properties. The scientific reports indicate that low-molecular-weight esters (LME) (triethyl citrate and bis (2-ethylhexyl) adipate) affect the plasticization of PLA. However, the research is limited to flat structures, such as films, casts, and extruded shapes. A study was conducted to investigate the impact of esters on the process of forming, the properties, and the morphology of fibers formed through the melt-spinning method. It was found that the modified PLA required different spinning and drawing conditions compared to the unmodified polymer. DSC, FTIR, WAXD, and GPC/SEC analyses were performed for the modified fibers. Mechanical tests and morphology evaluations using SEM microscopy were also conducted. The applied plasticizers lowered the temperature of the spinning process by 40 °C, and allowed us to obtain a higher degree of crystallinity and a better tenacity at a lower draw ratio. GPC/SEC analysis confirmed that the polymer–plasticizer interaction is physical because the booth plasticizer peaks were separated in the chromatographic columns. The use of LME in fibers significantly reduces the temperature of the spinning process, which reduces production costs. Additives significantly change the production process and the structure of the fiber depending on their rate, which may affect the properties, e.g., the rate of degradation. We can master the degree of crystallinity through the variable amount of LME. The degree of crystallization of the polymers has a significant influence on polymer application.

## 1. Introduction

Degradable polymers derived from natural sources are increasingly being used in a variety of industrial sectors like packaging, textiles, and medicine. This is due to the growing demand to replace petrochemical-based polymers with more eco-friendly alternatives such as polybutylene succinate, poly(lactic acid), poly(3-hydroxybutyrate), and polycaprolactone [1,2]. Medicine is one of the key sectors of biopolymer usage. This is due to their desired properties of gloss, transparency, antibacterial activity and flame retardancy [3,4,5]. The most commonly used biopolymers in medical materials are polysaccharides like chitosan, alginate, hyaluronic acid, etc.; proteins like collagen, silk, fibroin, etc.; and synthetic polymers like poly(glycolic acid), poly(lactic acid), poly-ε-caprolactone [6,7,8,9,10]. Unfortunately, their properties are sometimes a limitation to specific applications; therefore, modification processes are commonly carried out to obtain other desired features. Modifications could be chemical, physical, and biological, and aim to give new functional properties to medical materials. At present, the development in this particular direction is quite prevalent, mainly due to the high demand in the market. This demand arises from the progress of civilization, technology, as well as society. Given the aging population, there is a growing need for high-quality medical care. Thus, the progress in this technological area is of utmost importance. This problem concerns not only Europe but the entire world population. The forecast of the United Nations Organization predicts that by 2030, the percentage of European people over 65 years will reach 23.8%. The modern society, whose life expectancy is increasing expects new medical technologies that will enable people to stay healthy but also improve life quality and day-to-day functioning.

Polylactide (PLA) is a well-known biopolymer due to its origin, degradability, and processability. It can be processed at low temperatures and through various techniques including extrusion, injection molding, casting, blown film, thermoforming, and fiber spinning [11,12]. PLA is highly suitable for biomedical applications due to its biocompatibility. The degradation products of PLA are non-toxic and do not interfere with tissue healing. When PLA is hydrolyzed in the human body, it produces α-hydroxy acid, which is then incorporated into the tricarboxylic acid cycle and excreted [13]. But this polymer also has some limitations like poor toughness, brittleness and low flexibility. The poor toughness limits the use of PLA polymer in applications that require plastic deformation at higher stress levels, e.g., screws. The rate of in vivo degradation of PLA is slow. The research indicates that PLA’s lifetime could reach years in some cases; it mostly reaches the range of 3–5 years [14]. The degradation rate and mechanical properties are often considered important selection criteria for biomedical applications [15]. Scientists and the industrial sector find PLA interesting due to its natural origin and the trend of replacing petrochemical polymers with polymers. This interest results from the possibility of degradation and ease of processing with various additive manufacturing technologies, e.g., 3-D printing and composites [16,17,18].

Polylactic acid (PLA) is a versatile material commonly used in medicine. It can be produced using two methods: melt-spun technology, which is similar to the production of other melt-spun fibers like polypropylene, or solution spinning (both dry and wet) [19]. PLA is not only used for absorbable sutures but its fiber braid can also be used in human tissue repair and as a drug delivery system. During the material’s degradation process, pharmaceuticals can be released successively [20,21]. Additionally, the fiber can be used to create structures such as surgical meshes used in bone tissue engineering or as an Achilles tendon [22,23,24,25]. For the spinning process, the crystallization and degradation processes are important aspects, as they influence the mechanical properties. These aspects, however, determine the applicability of the material. Therefore, understanding the spinning process is crucial. In the medical field, knowledge of the mechanisms allows the development of a fiber material for an application. It is particularly important to be able to design medical fibers so that the modifier shows a dependence on a key parameter, e.g., strength, degradation time, elasticity, etc. The fibers for suture threads for dermal, hypodermic or muscular applications will require completely different requirements. Therefore, research into modifications and their effects on the properties of the material from a medical perspective is important.

As part of the research, a method of producing fibers with the addition of plasticizers was developed, which will allow for the obtainment of material with modified properties. There have been numerous studies conducted on modifying PLA to decrease its melt viscosity, increase its ductility, and improve its processing properties. The presence of a plasticizer in polymers influences the processes of crystallization and melting, lowering not only the *T_g_*, but also, to a lesser extent, the melting point of the polymer crystals (*T_m_*) [26,27,28,29,30,31,32,33,34,35,36]. There is no available information in the literature about the impact of low-molecular-weight ester compounds on the spinning process of fibers and their properties. However, these compounds affect the temperature of phase transformations and greatly influence the primary and secondary crystallization process of the fiber. The properties of fiber, such as its strength, elongation, and susceptibility to degradation, can be reflected in the final product made from it. By modifying these properties, it is possible to create new materials that can be used in a variety of ways. This is particularly interesting in the medical field, where fibrous materials with modified properties can have important applications. Therefore, it is important to determine what changes are caused by the plasticizers in the fibrous materials.

The objective of the research was to create a fiber technology by adding an additive that can control the crystallization process, resulting in a material with adjustable thermal and mechanical properties. By using various types and concentrations of additives, we can regulate the crystallization process and design materials with diverse properties. It is possible that new application sectors could be opened up for the use of PLA material. This is because PLA is a polymer with high application capabilities. To improve the plastic properties of PLA, it was modified using plasticizers, which are low-molecular-weight esters (LME) such as triethyl citrate (TEC) and adipate (ADO). These plasticizers reduce the interaction of polymer chains, thereby enhancing the plastic properties of PLA [26,37]. LME compounds were chosen because the introduction of small molecules with a *T_g_* value lower than the polymer *T_g_* causes an increase in the free volume of the composite polymer LME and an increase in the number of end groups according to the free volume theory. They enhance the mobility of polymer molecules and thus increase the flexibility and decrease polymer *T_g_* [38,39]. Numerous reference sources in the literature have confirmed the plasticizing effect of certain materials. A recent study was conducted to examine the properties of materials in the form of films and extrusion fittings. The results showed a significant improvement in the plastic properties of both the film and fittings [26,31,40]. There is currently no available information on the impact of using plasticizers on the structure and properties of fibers produced during spinning. Previous studies have shown that these two compounds have an intriguing effect on the properties of granulated polylactic acid (PLA) [41]. Therefore, TEC and ADO were selected to investigate whether the fibrous structure undergoes a polymer plasticization process. In the course of this paper, research to determine the influence of plasticizers on the thermal and mechanical properties of the fibers was conducted. The aim of the research was to study how LME affects the spinning process and the structure of fibers. In the process of melt spinning, the function of additives plays an important role in determining the degree of crystallinity. This influences the mechanical properties and susceptibility to degradation of the fibers. To better understand the polymer–plasticizer composition, it is crucial to determine the type of interaction taking place between them. In this research, regranulation with TEC and ADO was conducted in the range of 5–14% wt. Fibers were then produced from the regranulates by using the melting method. Modified fibers were analyzed using DSC, GPC/SEC, SEM, FTIR, and WAXD, and their mechanical parameters were evaluated for effectiveness.

## 2. Materials and Methods

### 2.1. Raw Materials

Polymer poly (D, L-lactide) 6201D used for the research came from Nature Works^®^ LLC (Plymouth, MN, USA). PLA was modified by two low molecular esters (LME): triethyl citrate and bis (2-ethylhexyl) adipate. Substances used in the study differed in molar mass and structure (Table 1).

### 2.2. Technological Processes

#### 2.2.1. Modification of Polymer

The first step the was production of regranulates by mechanical mixing of polymer containing 5–14% wt. of the low molecular esters in the molten state in a twin-screw extruder (Zamak Mercator, Skawina, Poland), equipped with eleven heating zones operating in the temperature range of 160–220 °C. Extrusion was carried out at a screw speed of 150 rpm and torque in the range of 0.5–2 Nm. A single-orifice forming head was used. After extrusion, the filament was quenched in a water bath and then cut into pellets, which were first air-dried at an ambient temperature of approx. 23 °C and then put for 48 h in the vacuum dryer (Binder, Tuttlingen, Germany) at 35 °C under reduced pressure (20 hPa).

#### 2.2.2. Spinning Process

The fibers were formed by a two-step melt-spinning process in a two-step process, comprising spinning and drawing. A multifilament fiber was spun on an experimental extruder spinning bank at the melting temperature range of 180–230 °C for raw PLA and 140–190 °C for modified polymers with a capacity of 28 g/min and speed of 900 m/min through a 24-hole spinneret (D = 0.3 mm, L = 1.0 mm). Estesol PF790 (Bozzetto Group, Filago, Italy), in a 13% wt. solution, was used as the spin finish. The spun fibers were drawn, using the SZ-16 (Barmag, Remscheid, Germany) draw–twister; the parameters of the process are presented in Table 2.

### 2.3. Methods

The thermal analysis was carried out by means of differential scanning calorimetry (DSC) using Diamond (Perkin Elmer, Waltham, MA, USA). The first and second heating scan and the first cooling scan for the polymer were performed in a temperature range of −60 ÷ 200 °C. The samples were scanned at a heating rate of 20 °C/min. The degree of crystallinity (*X_c_*) was calculated by comparing the difference between the melting and crystallization enthalpies with the melting enthalpy of a 100% crystalline PLA sample from the equation:(1)$$X_{C}=\frac{{∆H}_{m}-{∆H}_{CC}}{{∆H{}^{\circ}}_{m}}\cdot 100\%,\;[\%]$$
where

∆*H_m_*—the melting enthalpy (first heating of the sample);

∆*H_cc_*—the enthalpy of cold crystallization;

∆*H°_m_*—the melting enthalpy of 100% crystalline PLA (93.1 J/g) [42,43].

The melt flow rate (MFR) estimate was established according to the authors’ own methodology using an LMI 4003 plastometer (DYNISCO Polymer Test, Rochester, NY, USA). The polymer sample melted at 180 °C was extruded through a spinneret with a 0.5 mm capillary, at a piston load of 2.16 kg. The weight of the polymer extruded in the defined time (10 min) was determined, and the MFR was calculated. The filament obtained during the analysis was used for mechanical tests to assess the influence of the plasticizers on the mechanical properties of PLA.

Molar mass distribution and the polydispersity of polymers were analyzed by the gel permeation chromatography/size exclusion chromatography (GPC/SEC) method. The tests using Agilent Series (Agilent Technologies, Santa Clara, CA, USA) equipped with the Optilab refractometric detector (Wyatt Technology, Goleta, CA, USA) were performed. The tests were performed using chloroform HPLC as the eluent and one PL gel Mixed-C 300 mm chromatographic column (Agilent Technologies, Santa Clara, CA, USA) at a flow speed of 0.7 cm^3^/min. The average molar weights were determined using the universal calibration technique and the following values of the Mark–Houwink–Sakurada equation: for polystyrene (PS) a = 0.794, K = 0.0049 [44] and for PLA a = 0.759, K = 0.0153 [45].

Mechanical properties of fibers were determined under standard environmental conditions of 20 ± 2 °C and RH = 65 ± 4%, according to the PN-EN ISO 139:2006 standard [46]. The mechanical parameters such as linear density, breaking tenacity, and elongation at break were estimated using the PN-EN ISO 2062:2010 method A [47] on an Instron 5544 Tensile Tester (Norwood, MA, USA).

The structural analysis of fibers was inspected by means of a scanning electron microscope (SEM) Quanta 200 (FEI, Eindhoven, The Netherlands). Samples covered with a 20 nm layer of gold were tested in a high vacuum at an electron-beam-accelerating voltage of 10 kV.

Functional groups in fiber samples were determined by Fourier Transform Infrared Spectroscopy (FTIR) using a Nicolet iS50 Spectrometer (Thermo Scientific, Waltham, MA, USA) by transmission technique. The operating parameters were as follows: measurement range 4000–400 cm^−1^, resolution 4.0 cm^−1^, the number of scans for baseline and spectrum collection, and the accuracy of wavenumbers reading for characteristic bands was ±1 cm^−1^.

Wide-angle X-ray diffraction (WAXD): The crystalline structure of fibers was characterized using an X’Pert PRO diffractometer (PANalytical, Almelo, The Netherlands). Diffraction patterns of powdered samples were recorded within the range of angles 2θ: 5°–60° by using a CuKα (λ = 0.154 nm) X-ray source and the following tube parameters: accelerating voltage 40 kV, anode current density 30 mA.

## 3. Results

### 3.1. Analysis of Modified PLA Regranulates

For modified regranulates, rheological (Figure 1), thermal (Table 3) and molecular (GPC/SEC) (Figure 2 and Figure 3) analyses were performed.

The results show that the MFR parameter increased linearly with the concentration of modifiers, and the coefficient of determination R^2^ was high. By using the linear function equation, it is possible to produce a modified PLA regranulate with a specific MFR coefficient value. The glass transition temperature for the modified samples increased as the content of the modifier (TEC or ADO) was decreased (Table 3). A single glass transition was observed for the modified regranulates, indicating that the prepared compositions are miscible. However, during the second heating, a double transition to the liquid state was observed, with two *T_m_* values. This was caused by the presence of an additive that affects the polymer’s melting process (Table 3). Based on the free volume theory, plasticizers (modifiers) act on the amorphous part of the polymer, reducing the interaction between polymer molecules. The analyses conducted have confirmed the theory, indicating an increase in MFR and a decrease in *T_g_*. Additionally, the presence of modifiers allows for faster flow and movement of polymer molecules towards each other during the transition to a liquid state, due to increased flexibility in the polymer chains. The presence of the modifier in the crystalline phase may have caused double *T_m_* values as it flowed only in the melting point range. In the case of melting, the presence of the modifier may allow the occurrence of the polymorphism. A substance can appear in more than one crystal form which differs in physicochemical properties and can coexist simultaneously. In the case of a polymorphic substance, according to the Ziabicki model, it can occur in the following phases: liquid (amorphous) and in various crystalline forms, where phase transitions are possible. However, this aspect will be developed later in this article after DSC and WAXD fibers analysis.

GPC/SEC analysis was conducted on the modified regranulates to confirm the reason behind the increase in MFR, which was reduced polymer interactions and lower viscosity instead of degradation. For samples containing a modifier in the range of 5–7% wt., the differences in the value of the molar mass (*M_w_*) to the value of the base sample were equal to approx. 6% (Table 3). For higher concentrations, the differences were insignificant. Additionally, in the graph of the molar mass distribution, no significant differences were observed (Figure 2 and Figure 3). During the GPC/SEC analysis, apart from the polymer peak, the peaks of TEC and ADO on the chromatogram were observed, which shows that it was a physical modification. The concentrations of TEC and ADO were determined in the regranulate by using the modifier peak area and the calibration curve. The obtained values for ADO were the same as those assumed. However, for TEC, the values were slightly lower (Table 3).

### 3.2. Analysis of Modified Fibers

After the spinning and drawing process, two types of fibers were obtained. One type was made of pure PLA with a draw ratio (dr) of 3.35, while the other type had modifiers with variable content and draw ratios of 2.80 and 3.28. The presence of modifiers reduced the spinning temperature by 40 °C. However, the modified fibers had a lower draw ratio than the pure PLA fibers because they tended to break during the draw–twisting process.

The designation of fibers:Base fibers from pure PLA dr = 3.35;Fibers with TEC at a concentration of 5, 7, 12% wt., two draw ratios—2.80 and 3.28;Fibers with ADO at a concentration of 5, 7% wt., two draw ratios—2.80 and 3.28.

As a result of spinning, it was not possible to obtain fibers with ester plasticizers in the amount exceeding 10% wt. (only for TEC 12% wt. fibers were obtained) because the extruder did not take up the plasticized regranulates. During spinning, the extruder did not move the material to the filing system. The modified material showed too low an MFR value, and the screw did not take up the material, so it was not possible to obtain the pressure that would guarantee the proper spinning.

#### 3.2.1. DSC Analysis of Modified Fibers

According to the DSC analysis, the glass transition temperature of the fibers was found to be similar to that of citrate-modified regranulates (TEC) as shown in Table 4. However, for fibers with ADO, the *T_g_* value was slightly higher than that of regranulates. Moreover, a double liquid transition into the plastic state for this compound was also observed, because two *T_m_* values were evidenced (double peak). In base granules and fibers PLA this phenomenon did not exist (Figure 4, Table 3). After modification, it was observed for regranulates with TEC and ADO (Table 3). The bimodal peak of mealing for fibers with ADO (Figure 5, Table 4) after spinning and also after drawing was observed. In the case of TEC, this phenomenon was only noticeable in the regranulate, and in the second heating for fibers after and before drawing. (Figure 6 and Figure 7). During the first heating, the DSC analysis was applied to the form of the material, i.e., fibers; in the second heating, the material was examined. For fibers with TEC before and after the first heating, a double peak of melting was not observed. Unfortunately, the scientific literature explains such a phenomenon in different ways. During the fiber orientation process, a change in the crystal structure may cause the melting behavior to become more complicated, with two observed melting temperatures (*T_m_*_1_ and *T_m_*_2_). Depending on processing conditions, PLA can crystallize in three forms [48,49,50,51]. During melting or in cold conditions, crystallization occurs in the α form [52]. β form crystallization occurs by stretching the α form or by solution-spinning processes conducted at high temperatures or high hot–draw ratios [49,53]. The third form called γ was obtained by Lotz and co-workers via epitaxial crystallization on z hexamethylbenzene (HMB) substrate [50]. Researchers have observed two melting temperatures, which are associated with links to the phase transition between a metastable crystalline α′-form to the stable crystalline α-form [52,54,55]. The research suggests that the presence of a double melting peak in PLA films is likely due to the formation of thinner and/or less perfect crystals during the cooling process after annealing. This phenomenon is only present in a small fraction of the crystals and contributes to the double peaks observed in the melting curve [35,56].

The modified fibers had a higher draw ratio, which resulted in an increase in the axial orientation and, hence, an increase in the crystalline phase, as shown in Figure 8. These modified fibers were stretched with a smaller draw ratio than the PLA fibers. After modification, all the variants (with a draw ratio of 3.28) had a higher crystallinity than pure PLA fibers (with a draw ratio of 3.35). For fibers with 7% wt. TEC for dr = 2.80, a higher crystallinity value was obtained than for PLA fibers and similar to the higher dr = 3.28, but for a lower TEC concentration of 5% wt. The addition of modifiers to the fibers increased their crystallinity but resulted in a lower draw ratio compared to PLA fibers. Based on the plasticization free volume theory, the interaction between TEC and ADO with the polymer occurs mainly in the amorphous phase. Their presence decreased interaction between the polymer chains, which resulted in changes in MFR and *T_g_* values. It was observed that a lower draw ratio led to more effective crystal formation. The presence of a plasticizer facilitated better disengagement, straightening, and paralleling of the macromolecules in amorphous regions, resulting in greater mobility of the amorphous phase and less tensile stress.

#### 3.2.2. GPC/SEC Analysis

The fibers were subjected to chromatographic analysis to determine the degree of polymer degradation as a result of the spinning process. The results are presented in Table 5; the relative deviation standard value for the results did not exceed 3%. The spinning process, apart from the regranulate with 12% wt. TEC caused a slight change in the value of the molar mass of the fibers to the regranulates from which they were made. Furthermore, the difference in the molar mass of the modified fibers concerning the PLA base fiber was small. The GPC/SEC method allows for a 5% percentage difference in the results, therefore it can be concluded that the spinning process was conducted under optimal conditions. It was also not observed that the draw ratio affected the molecular parameters. Slightly lower amounts of modifiers after the spinning process were observed. The modifier peaks in the GPC/SEC analysis were distinguished; thus, this proves that it was a physical interaction between the polymer and plasticizer.

#### 3.2.3. Mechanical Properties

The modified fibers were subjected to mechanical analysis, the results are presented in Figure 9, Figure 10 and Figure 11. Figure 9 shows that the linear mass of the modified fibers is slightly higher for a draw ratio (dr) of 2.80 and similar for dr = 3.28. The low measurement uncertainty values confirmed the achievement of a regular fiber structure. Despite a lower dr, the modified fibers exhibited greater breaking tenacity, as shown in Figure 10. In the stretching process, molecules in the amorphous region and existing crystallites, along with those formed by orientation crystallization, were ordered. For the base fiber, the elongation of about 50% for dr = 3.35 was obtained. For the modified ones, a similar value for dr = 2.80 was obtained (Figure 11), where the higher breaking tenacity was determined. The lower elongation at break of approx. 25% for modified fibers for dr= 3.28 was determined. Parameters indicating increased strength for the modification with ADO were obtained.

#### 3.2.4. SEM Analysis

The modified PLA fibers underwent morphological evaluation through scanning electron microscopy. The resulting observations showed that the modified fibers maintained their correct morphology, as indicated in Table 6. For all samples, the fiber structure remained undisturbed after the modification process. In Table 6, exemplary SEM images for the modification with 7% wt. TEC and ADO were presented. This reveals that optimal conditions for the spinning process have been developed. The obtained fiber diameters for the basic one were similar, which after modification with dr = 2.8 (approx. 20 µm), for dr = 3.28, were slightly lower (approx. 19 µm) (Table 7). The relationship between fiber diameter and draw ratio (dr) is also demonstrated in the fiber line mass.

#### 3.2.5. Spectral Analysis

For PLA fibers with 7% wt. triethyl citrate (TEC) and bis (2-ethylhexyl) adipate (ADO) (dr = 3.28) structural analyses were performed using the FTIR technique (Figure 12, Figure 13, Figure 14 and Figure 15).

The difference in the high wavenumber region of 3100–2600 cm^−1^, characteristic of the C-H stretching vibration was observed only for ADO (Figure 12). Plasticization with ADO caused disturbances for bands around 2950 cm^−1^ (symmetric stretching of C-H in the methyl group-υsCH_3_) [57]. For TEC, this phenomenon was not observed [41]. The change in the region 3100–2600 cm^−1^ for ADO fibers may indicate an interaction of the modifier with the methyl group in PLA. The FTIR spectrum of the base PLA showed the low intensity of the band at 920 cm^−1^ and higher after all modifications, which can be linked to the coupling of C-C backbone stretching with the CH_3_ rocking mode and sensitivity to chain conformation of PLA α or α’crystals (Figure 14) [58,59,60]. Two bands related to crystallinity are also important: at 872 cm^−1^ (the amorphous phase) and 757 cm^−1^ (the crystalline phase in the PLA) [61,62]. The amorphous phase is more accessible to plasticizers; therefore, the higher band intensity was observed for fibers with ADO, as well as for the crystalline region (757 cm^−1^) for modifications. The impact of the modifications in the amorphous part was confirmed by FTIR analysis. Additionally, the increase in the degree of crystallinity was confirmed to occur due to the modification. The bands at 1630 cm^−1^ and 1570 cm^−1^ were observed for the researched samples (Figure 13). The publications indicated that, at approx. 1600 cm^−1^, a band appears after degradation and is assigned to the carboxyl ion (–COO^−^), formed as a result of the hydrolysis of the ester group [63,64]. GPC/SEC analysis confirmed that the degradation process had occurred. The Fourier self-deconvolution process of the carbonyl group band was performed (Figure 15). The results were optimized by keeping the bandwidth (the value = 117.0), and enhancement parameters (the value = 2.0) value the same for all samples. The bandwidth is an estimate of the widths of the overlapped bands. Enhancement is a measure of the degree to which features are revealed. It determines the “strength” of the resolving power applied to the data. A difference was observed in the band deconvolution plot (FSD plot) for the modification with ADO. Similar results were obtained with the FTIR-ATR technique in other examinations when a band was deformed [41]. This band for fibers modified with ADO is more complex, which may indicate an interaction of the carbonyl group of polymer with the plasticizer. This region (1700–1800 cm^−1^) is sensitive to the conformation at the chain and it can be used to differentiate the crystalline form or crystals of PLA.

#### 3.2.6. WAXD Analysis

For base and modified PLA fibers WAXD analyses were performed (Figure 16). The aim of the study was to determine if the double melting peak is due to different crystalline phases or thinner and/or less perfect crystals in accordance with two earlier theses presented by other researchers. The examination demonstrated no differences in crystalline phases for the base and modified fibers. For modified fibers, the temperature of the spinning process was significantly reduced. According to Avarami’s theory, crystallization is the resultant of two processes that occur simultaneously, nucleation—the formation of crystal nuclei and aggregation, which is the process of crystal growth. Both processes are dependent on temperature but their intensity at the same temperature is different. Figure 17) [65]. Because the modification shifted the crystallization curve toward nucleation, it resulted in a fine-grained structure and the crystals were thinner and/or less perfect. The effect of this shift in crystallization toward intense nucleation was a double melting peak and changes visible on FTIR in region 1700–1800 cm^−1^ and on the Fourier self-deconvolution plots of the carbonyl group band.

In Figure 16a, the WAXD profiles of the obtained fiber are presented. On the diffractograms of base and modified PLA fibers, three dominant diffraction peaks located at 2θ 16.5°, 18.8° and 28.8° are clearly visible, corresponding to (110)/(200), (203) and (216) lattice planes of α’ forms of PLA. Additionally, the insignificant peak located around 2θ 22.3° corresponding to (015) lattice planes of polylactide crystalline forms is visible for the fibers containing 7% of additives ADO and TEC. Based on the presented WAXD diffraction profiles, it may be concluded that, according to DSC studies, ADO and TEC additives promote PLA crystallization during the fiber-forming process.

The more detailed structural analysis of PLA fibers was obtained by the deconvolution of the X-ray diffraction profiles into the amorphous halo and the crystalline peaks. For this analysis, the experimental data were fitted by a composite of the Gauss and Lorentz functions calculated using the WAXSFIT software 1.0 based on Hindeleh and Johnson’s method [66,67]. The shapes of the amorphous halo and the mesomorphic and crystalline peaks were selected according to the model proposed by Stoclet et al. [68]. An example of the deconvolution of the diffraction pattern obtained for the base PLA fibers is presented in Figure 16a. The crystalline and mesomorphic phase contents in the studied materials were calculated according to the following equation:(2)$$\chi_{O}=\frac{A_{C}+A_{M}}{A_{A}+A_{C}+A_{M}}\cdot 100\%$$
where *A_A_*, *A_C_* and *A_M_* are the integral intensities of the amorphous halo and the peaks originating from the crystalline phase and meso-phase, respectively. Additionally, the size of crystallites perpendicular to the lattice planes (110)/(200) was calculated by measuring the FWHM of the most visible diffraction peak using Scherrer’s formula:(3)$$L_{\left(hkl\right)}=\frac{K\lambda }{B\cos\theta }$$
where *L_(hkl)_*—average size of crystallites perpendicular to lattice planes (hkl); *θ*—Bragg angle for planes (hkl); *λ*—wavelength of X-ray radiation (for CuKα *λ* = 0.154 nm); B—FWHM of the diffraction peak for planes (hkl); *K*—Scherrer’s constant that, for the polymer, is equal to 1.

In Figure 16b, changes in the crystalline structures are presented. It is clearly shown that the addition of ADO and TEC slightly increases the crystallinity of the obtained fibers. The meso-phase is still observed, which is typical for the α’ forms of PLA-ordered structures which also allow a high tenacity of fibers. The presented meso-phase also favorably affects breaking elongation, and no brittleness effect is observed, which is evident for fibers containing a crystalline structure of α form of PLA [69]. Additionally, the influence of ADO or TEC content on the increase in meso-phase content was observed and this supports the hypothesis of additive incorporation into the fiber structure in not only amorphous but also mesomorphic areas. Moreover, the analysis of the changes in crystalline size in the function of additives content was conducted. The presented in Figure 16b results suggest different crystallization mechanisms in the case of oriented PLA fibers with the addition of Ado or TEC than in base fibers. The incorporation of molecular additives provides nucleation and the creation of fine-grained structures, but in the case of base fibers, the aggregation process is dominated. The WAXD technique allows obtaining the average structural information, while DSC thermograms contain two melting points that testify about the existence of two various sizes of crystallites of the α’ form of PLA. The different crystallization mechanisms and different sizes of crystalline areas explain mechanical properties; in this case, a decrease in elongation with an increase in tensile strength. WAXD analysis is therefore one of the keys in interpreting the results of mechanical testing of fibers. Mustapa et al. have shown that TBC (similar to TEC) decreased the concentration and increased the sizes of spherulites ascribed to the plasticization effect of TBC w PLA [61,70]. Murariu et al. has shown that bis(2-ethylhexyl) adipate improved flexibility and crystallization rates, and tributyl O-acetylcitrate (TBAC) improved elongation at break and tensile strength properties. The tests were conducted on extruded shapes [62].

## 4. Conclusions

Significant changes were observed in the polymer after the modification, including alterations in the melt flow rate and glass transition temperature. As a result, new optimal conditions were required for spinning and drawing the fibers. The modification necessitated a reduction in the process temperature by 40 °C in all heating zones during the spinning process, along with a lowering of the draw ratio. It was observed that the addition of modifiers increased the crystallinity of fibers that had a draw ratio lower than that of pure PLA fibers. The plasticization volume theory suggested that TEC and ADO interacted with the polymer mainly in the amorphous part, which was confirmed by FTIR analysis as indicated by a higher intensity of the band for the amorphous phase. Additionally, the presence of modifiers reduced the interactions between the polymer chains, which caused changes in MFR and *T_g_* values.

As a result of the research, it can be concluded that the use of a lower draw ratio resulted in the observation of more effective indicative crystallization. The greater mobility of the amorphous portion resulted in better plunge, straightening, and parallelization of the macromolecules in the amorphous regions where a plasticizer was present, with less tensile elongation. The isolated peaks of modifiers in the GPC/SEC analysis proved that the modification was physical. Better mechanical parameters were achieved with reduced fiber stretching times for modified fibers. The microscopic morphological analysis showed no structural disturbances, indicating that the process was conducted under optimal conditions. The molar masses of the fibers with 5 and 7% wt. of modifiers were determined at a similar level as regranulates. This was due to the decrease in temperature during spinning, resulting from slower polymer degradation. Further research is needed to determine the effect of esters on polymer breakdown.

The fibers undergo various processes, one of which is axial drawing which leads to further crystallization. The results obtained for the fibers were different from those reported in the literature for other forms such as films and extruder molds. The addition of ADO to the fibers helped to enhance crystallinity and increase breaking tenacity.

Improved mechanical properties were observed only in fibers modified with ADO, as evidenced by a difference in the band deconvolution plot in spectral analysis (FSD plot). The observed double melting peak for fibers with ADO on the DSC analysis excludes the explanation of different crystalline phases. In the spectra between PLA and ADO-modified fibers, no differences were observed. The phenomenon was most probably connected with the presence of a minute fraction of thinner and/or less perfect crystals formed during spinning [35,56]. As the presence of the plasticizer significantly reduced the process temperature, smaller and less perfect crystals could be formed during the crystallization process. Because the modification shifted the crystallization curve toward nucleation, it resulted in a fine-grained structure and the crystals were thinner and/or less perfect. The effect of this shift in crystallization toward intense nucleation was a double melting peak and changes visible on FTIR in region 1700–1800 cm^−1^ and on the Fourier self-deconvolution plots of the carbonyl group band. The WAXD technique allows us to obtain average structural information while DSC thermograms contain two melting points that testify about the existence of two various sizes of crystallites of the α’ form of PLA. The different crystallization mechanisms and different sizes of crystalline areas explain mechanical properties; in this case, a decrease in elongation with an increase in tensile strength. WAXD analysis is therefore one of the keys in interpreting the results of mechanical testing of fibers.

During the spinning process, the additives TEC and ADO were found to enhance the organization of the fiber structure. This effect was not observed in other materials such as film. It is important to note that the spinning process is unique and cannot be compared to any other process. Therefore, studying the impact of LME on the spinning process is a fascinating topic to explore. Additives significantly change the production process and the structure of the fiber depending on the amount, which may affect the properties, e.g., the rate of degradation. We can influence the degree of crystallinity through the variable amount of LME. The degree of crystallization of the polymers has a significant influence on the utility properties of polymers. The studies are continued and are aimed at determining the dynamics of hydrolytic degradation in simulated body fluids and biodegradation in compost. Initial research is very promising. Finding a solution that will allow you to control degradation will allow you to design products with a controlled time. Such materials can be used in medical engineering, for the design of surgical meshes, threads, and implants with a controlled degradation time, which will allow the body to recover properly. Also, in the aspect of agricultural materials, i.e., non-woven fabrics, strings, geo-pads, and pots, it will be possible to design the degradation time according to the needs, i.e., plant growth, protection period, and period of use.

## Figures and Tables

**Figure 1 materials-17-01268-f001:**
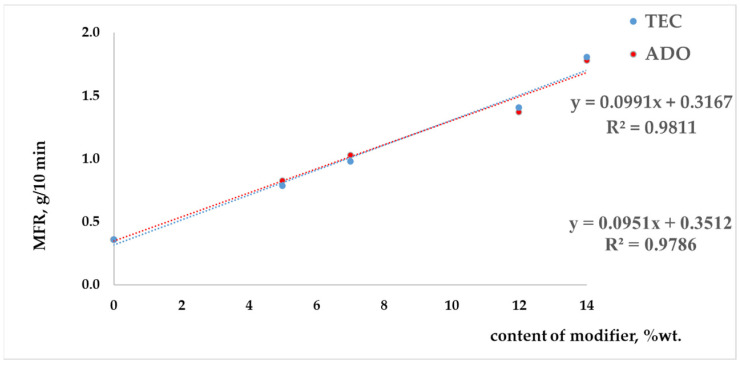
Melt flow rate for modified regranulates at 180 °C.

**Figure 2 materials-17-01268-f002:**
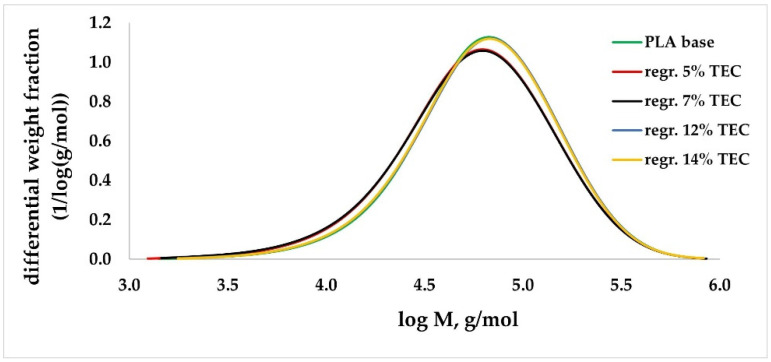
Molar mass distribution (MMD) for PLA with TEC.

**Figure 3 materials-17-01268-f003:**
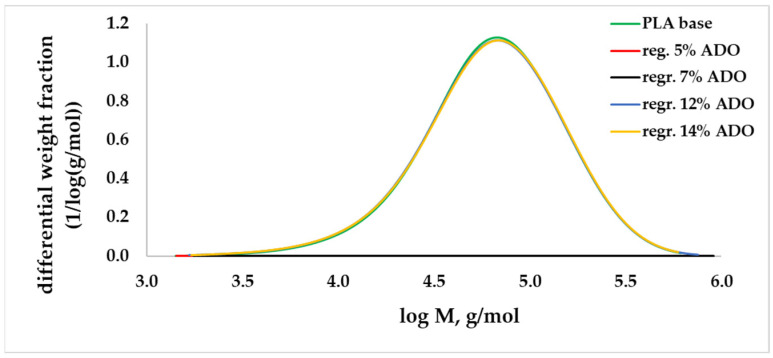
Molar mass distribution (MMD) for PLA with ADO.

**Figure 4 materials-17-01268-f004:**
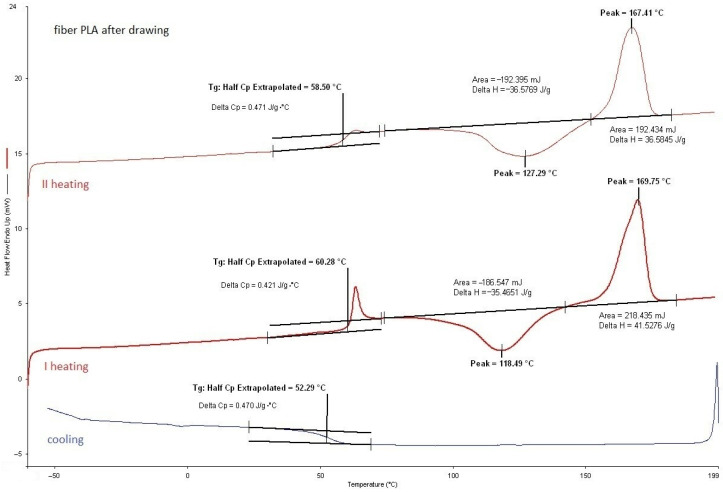
PLA fibers after drawing.

**Figure 5 materials-17-01268-f005:**
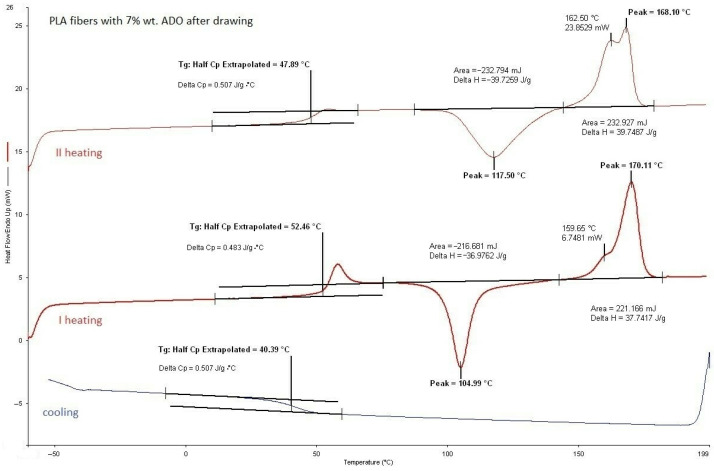
PLA fibers with 7% wt. ADO after drawing.

**Figure 6 materials-17-01268-f006:**
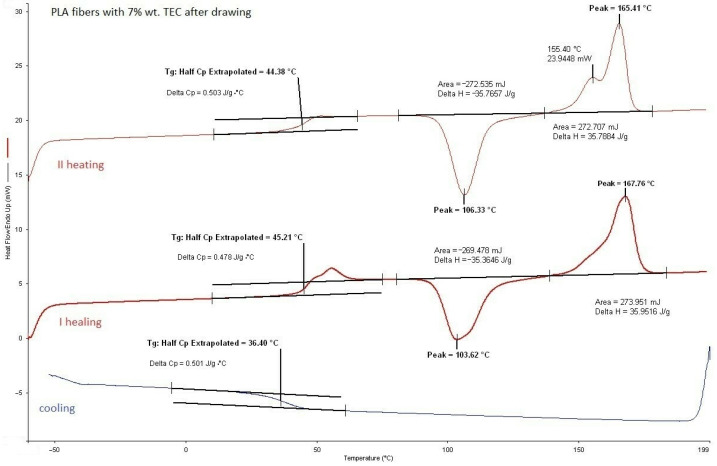
PLA fibers with 7% wt. TEC after drawing.

**Figure 7 materials-17-01268-f007:**
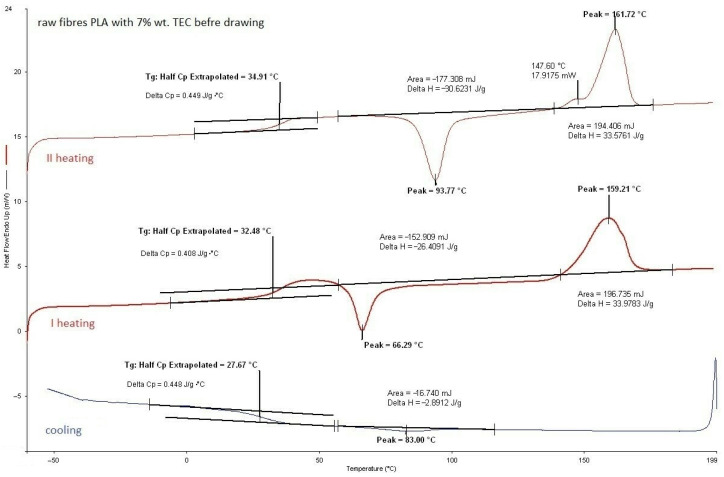
PLA fibers with 7% wt. TEC before drawing.

**Figure 8 materials-17-01268-f008:**
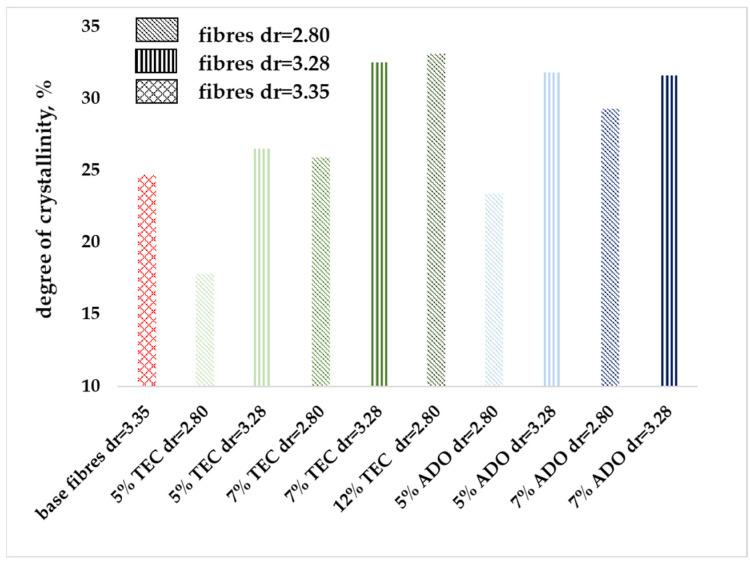
Degree of crystallinity for modified fibers.

**Figure 9 materials-17-01268-f009:**
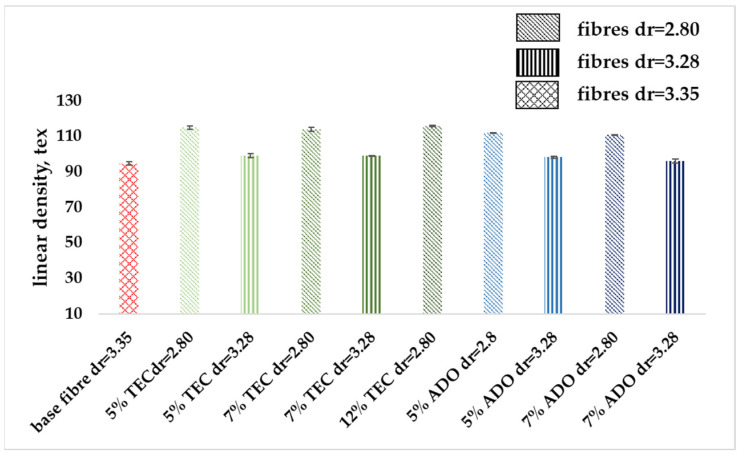
Linear density of modified fibers.

**Figure 10 materials-17-01268-f010:**
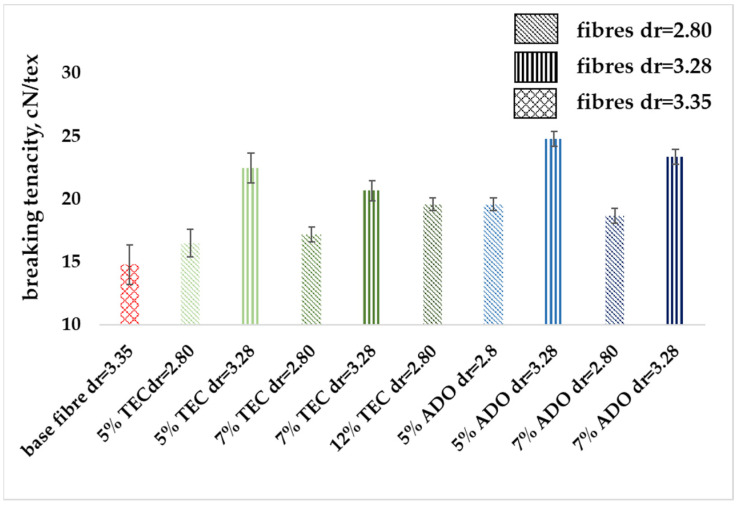
Breaking tenacity of modified fibers.

**Figure 11 materials-17-01268-f011:**
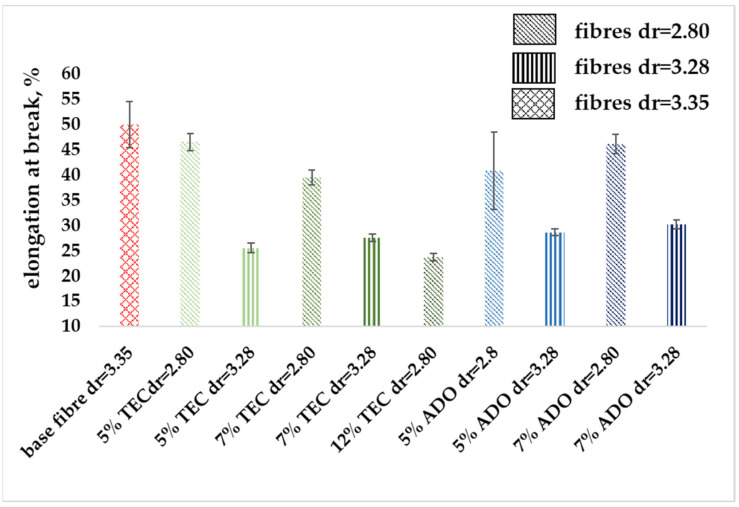
Elongation at the break for modified fibers.

**Figure 12 materials-17-01268-f012:**
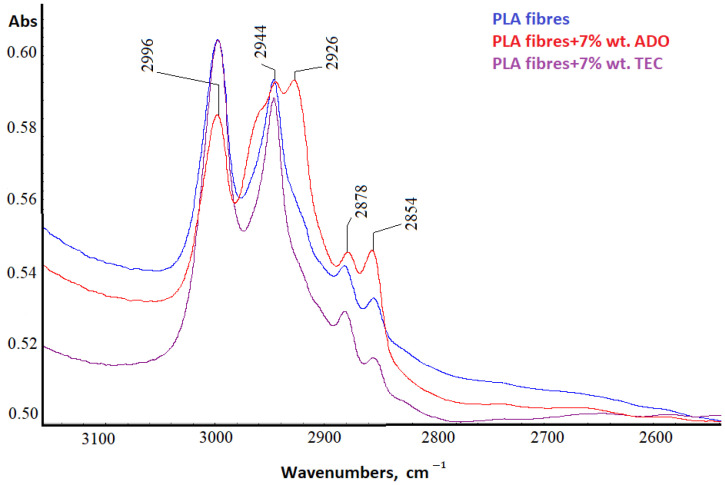
FTIR spectra in the range 3100–2600 cm^−1^.

**Figure 13 materials-17-01268-f013:**
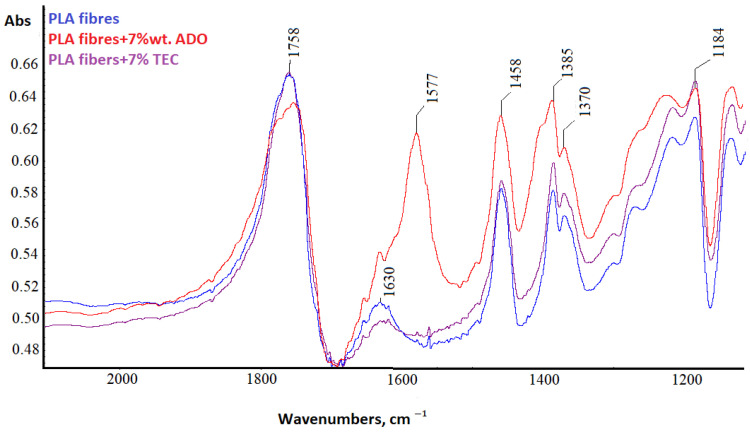
FTIR spectra in the range 2000–1200 cm^−1^.

**Figure 14 materials-17-01268-f014:**
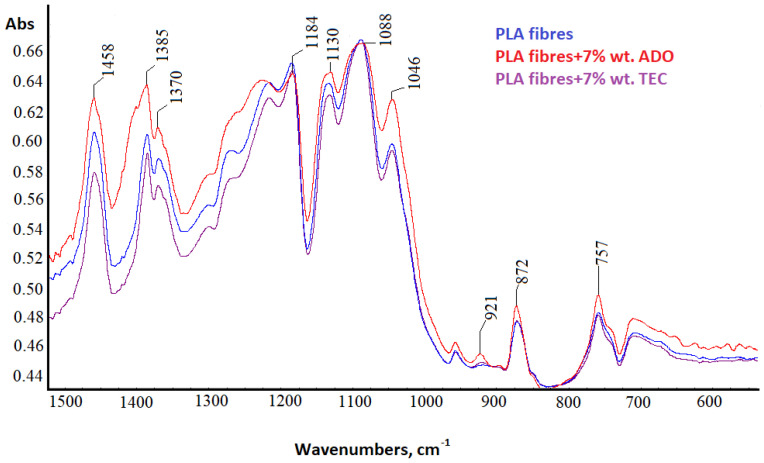
FTIR spectra in the range 1500–600 cm^−1^.

**Figure 15 materials-17-01268-f015:**
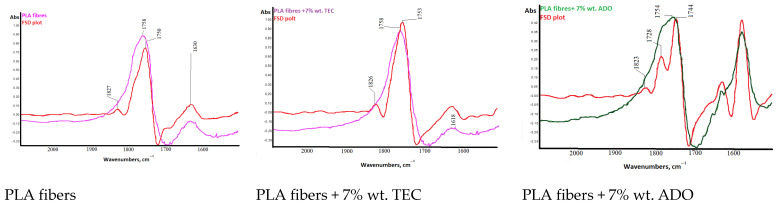
Deconvolution of band 1730 cm^−1^ for the ester group.

**Figure 16 materials-17-01268-f016:**
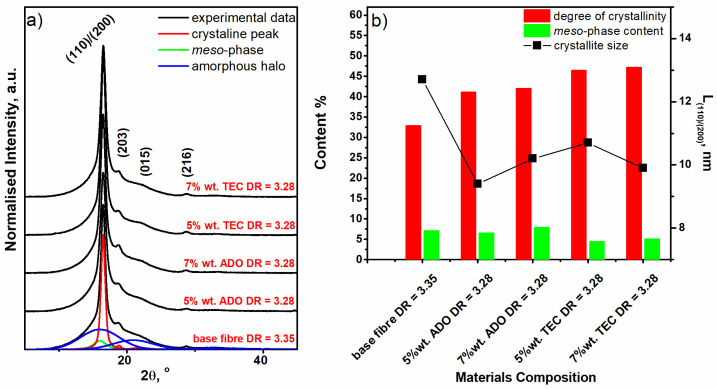
X-ray diffraction profiles of studied fibers with deconvolution of base PLA fibers (**a**) and crystal and meso-phase contents as a function of materials compositions (**b**).

**Figure 17 materials-17-01268-f017:**
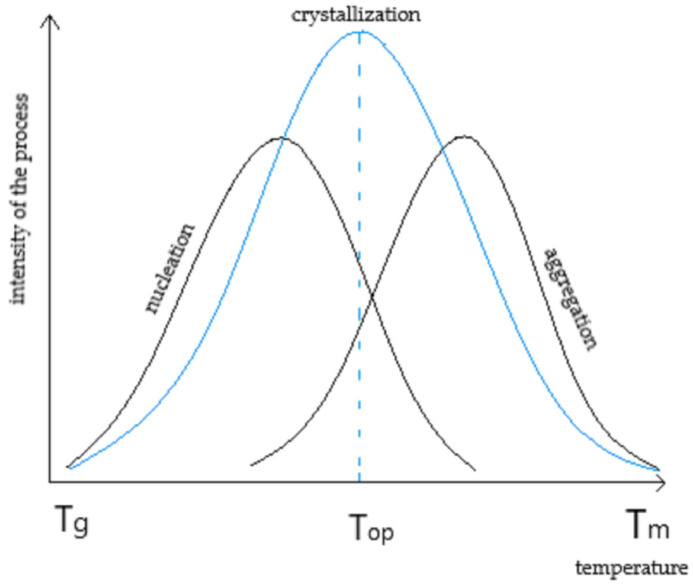
The crystallization curve [61].

**Table 1 materials-17-01268-t001:** Characteristics of studied substrates.

Substrates	Producer	*M_w_*, g/mol	Structure	Contents of D-lactde (%)
poly (D, L-lactide)	Nature Works^®^ LLC, Plymouth, MN, USA	82,700	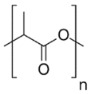	1.4
triethyl citrate (TEC)	Gentham Life Sciences Ltd, Corsham, UK	276	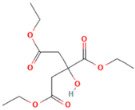	-
bis (2-ethylhexyl) adipate (ADO)	Boryszew S.A., Warszawa, PL, USA	370	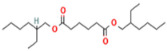	-

**Table 2 materials-17-01268-t002:** Parameters of drawing.

Sample	Draw Ratio (d_r_)	Temperature, °C		Spindle Speed, min^−1^ × 10^3^
		Goted	Plate	
Base fibers	3.35	70	120	6800
Modified fibers with 12% wt. TEC	2.80	60	110	7100
Modified fibers with 5% and 7% wt. of TEC	3.28	70	120	6600
	2.80	70	120	6848
Modified fibers with 5% and 7% wt. of ADO	3.28	70	120	6600
	2.80	70	120	6848

**Table 3 materials-17-01268-t003:** Parameters for modified regranulates.

Sample		DSC							GPC/SEC	
		*T_g_*, °C	∆*C_p_*, J/gK	*T_cc_*, °C	∆*H_cc_*, J/g	*T_m_*, °C	∆*H_m_*, J/g	*Χ_c_, %*	*M_w_*, g/mol	C_obtained_, %
PLA Base		60.0	0.50	123.9	42.1	165.9	42.2	0.11	82,700	-
	Modified Regranulates									
Modifier	C, % wt.									
TEC	5.0	49.2	0.50	110.2	37.4	158.0 166.6	37.4	<0.01	77,500	4.0
	7.0	44.4	0.50	106.3	35.8	155.4 165.4	35.8	<0.01	77,100	6.4
	12.0	34.6	0.49	98.0	34.4	147.1 161.4	34.4	<0.01	82,700	11.9
	14.0	30.1	0.47	96.3	33.2	145.0 159.5	33.2	<0.01	82,600	12.0
ADO	5.0	47.9	0.51	117.5	39.7	162.5 168.1	39.7	<0.01	77,200	5.0
	7.0	42.8	0.49	95.5	35.6	154.5 166.3	35.6	<0.01	77,600	7.0
	12.0	39.9	0.47	91.6	36.2	150.5 160.0	37.8	1.72	82,900	12.0
	14.0	38.3	0.49	91.1	35.1	150.8 162.6	37.1	2.15	83,300	14.0

**Table 4 materials-17-01268-t004:** Parameters for modified fibers from first heating.

Sample			DSC					
			*T_g_*, °C	∆*C_p_*, J/gK	*T_cc_*, °C	∆*H_cc_*, J/g	*T_m_*, °C	∆*H_m_*, J/g
Base Fibers PLA dr = 3.35			65.4	0.20	76.2	20.0	162.1	44.6
Modified Fibers								
Modifier	C,% wt.	dr						
TEC	5.0	2.80	48.5	0.29	65.1	16.1	161.2	42.5
		3.28	48.5	0.29	65.1	16.1	161.2	42.5
	7.0	2.80	44.5	0.26	61.3	15.5	161.4	41.4
		3.28	44.6	0.21	115.3	10.0	161.1	42.5
	12.0	2.80	33.2	0.14	112.6	6.2	158.2	39.4
ADO	5.0	2.80	49.8	0.33	62.9	20.1	154.2; 162.7	43.5
		3.28	51.0	0.20	78.4	13.4	153.9; 162.6	45.2
	7.0	2.80	46.5	0.21	71.2	13.6	155.3; 163.6	42.9
		3.28	49.0	0.18	110.9	11.8	154.8; 162.9	43.4

**Table 5 materials-17-01268-t005:** Parameters for modified fibers.

Sample			GPC/SEC					
			*M_n_*, g/mol	*M_w_*, g/mol	*M_w_/M_n_*	*M_w_* Difference to Granules, %		
Base Fiber PLA Rc = 3.35			38,000	78,700	2.1	4.9		
Modified Fibers								
Modifier	C, % wt.	dr	*M_n_*, g/mol	*M_w_*, g/mol	*M_w_/M_n_*	*M_w_* Difference, %		C_obtained_, % wt.
						To Regranulates	To Base Fiber	
TEC	5.0	2.80	35,100	76,600	2.2	1.1	2.7	4.0
		3.28	35,700	76,500	2.1	1.3	2.8	4.0
	7.0	2.80	36,000	76,400	2.1	0.9	2.9	6.6
		3.28	36,500	74,500	2.0	3.4	5.3	6.4
	12.0	2.80	33,500	70,400	2.1	14.9	10.5	10.6
ADO	5.0	2.80	36,200	76,600	2.1	0.7	2.6	4.4
		3.28	37,700	77,500	2.1	−0.4	1.5	4.7
	7.0	2.80	37,200	74,700	2.0	3.7	5.1	7.0
		3.28	37,200	75,900	2.0	2.1	3.5	7.0

**Table 6 materials-17-01268-t006:** SEM images.

Sample		500×	1000×
Base fibers PLA dr = 3.35		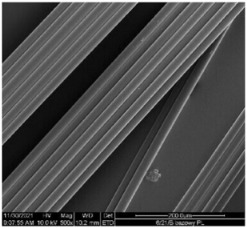	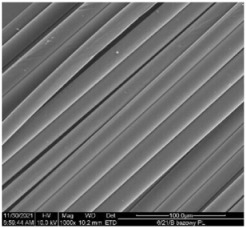
Modified fibers			
TEC, 7% wt.	dr = 3.28	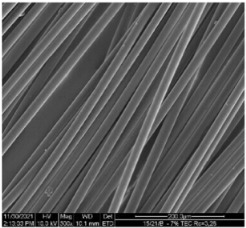	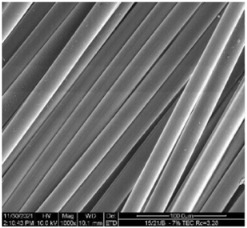
ADO, 7% wt.	dr = 3.28	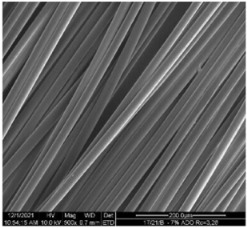	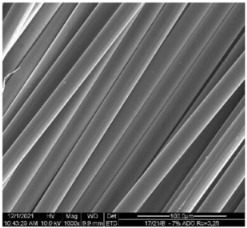

**Table 7 materials-17-01268-t007:** Fiber diameter calculated in the SEM program.

Sample			Fiber Diameter, µm	Linear Mass, dtex
Base Fibers PLA Rc = 3.35			20.3 ± 1.8	94.8 ± 0.6
Modifier	C, % wt.	dr		
TEC	5.0	2.80	21.9 ± 1.4	115.0 ± 1.0
		3.28	19.6± 0.9	99.2 ± 1.0
	7.0	2.80	21.1 ± 1.0	114.0 ± 1.0
		3.28	18.9 ± 1.4	99.0 ± 0.3
	12.0	2.80	21.5 ± 1.0	116.1 ± 1.3
ADO	5.0	2.80	20.8 ± 1.3	112.0 ± 0.1
		3.28	19.9 ± 1.4	98.3 ± 0.6
	7.0	2.80	20.5 ± 1.7	111.0 ± 0.2
		3.28	18.8 ± 1.5	96.1 ± 1.2

## Data Availability

Data are contained within the article.

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
