# Peer review of "Influence of Low-Molecular-Weight Esters on Melt Spinning and Structure of Poly(lactic acid) Fibers"

_materials, 2024, doi:10.3390/ma17061268_

Round 1
Reviewer 1 Report
Comments and Suggestions for Authors
This paper well describes the modification of PLA properties, oriented to fiber production, by the addition of two low molecular weight esters.
The target is clear, the introduction well referenced and the results/conclusions plausible.
Although the results could be of importance in the field of biomaterials, this paper reports simply one of the numerous examples of material modification by plasticization. The novelty in the materials science field is poor and, in my opinion, it would be more appropriately placed in a journal concerning polymer materials; I suggest Macromol of MDPI publisher.
The paper can’t be accepted in this version. It has to be strongly revised. Before publication it has to be submitted to revision to improve English language and to erase several errors. All the paper has to be more carefully curated.
Only SOME examples:
- Several acronyms are not explicated, please do it. In the abstract LEM must be indicated.
- LME or LEM ?? please uniform it along the entire manuscript.
- Pag. 2 line 68 dlimitationsfness ??
- Pag. 2 line 98 the line is not appropriately used
- Pag.3 line 101 ….with a lower Tg value than the polymer…. with a Tg value lower than the polymer….
- Pag. 3 line 106 The sentence ‘The properties of materials in the form of films, extrusion fittings were studied.’ seems truncated?
- the final part of Introduction at page 3 has to be rewritten in a better way.
- Table 1, the formulae of citrate and adipate are wrong, probably something happened during cut/paste process.
- Pag 5 line 191 a space is present in diffract ion
- Pag 6 line 210 a comma is necessary after the word ‘values’
- Pag 12 line 329 ….e… is not clear….??
- Pag 17 line 429 where is the equation? I can’t see it!!!
Figure 16 with WAXD Figure is not visible!!!!!!!!!! Did you put it?
...............
Comments on the Quality of English LanguageBefore publication it has to be submitted to revision to improve English language
Author Response
Hello,
thanks for reviewing my manuscript. All the comments and suggestions were very helpful and I began to use them. Thank you for your work and your time to read and improve my research work.
The new modifications as suggested by reviewers have been marked in red.
The language style has been improved.
Added in abstract:
The scientific reports indicate that low-molecular-weight esters (LME) (triethyl citrate and bis (2-ethylhexyl) adipate) affect the plasticization of PLA.
Corrected on LME.
All indicated mistakes and incorrect sentences have been corrected and highlighted in red.
Corrected: Keywords: poly (lactic acid) fibres; textile materials; triethyl citrate; bis (2-ethylhexyl) adipate; crystallization; supramolecular structure.
Inserted graphs that disappeared from the manuscript.
Reviewer 2 Report
Comments and Suggestions for Authors
The study attempted to give additional value to PLA through modification with low molecular weight ester, but the following observations need to be addressed before this manuscript can be reconsidered for publication by Materials:
1. In as much as possible, authors should avoid repeating part of title words as keywords.
2. Authors should specify which of the two stated standard methods that was used in mechanical properties evaluation.
3. In line 186, 'spectra analysis was carried out' should be changed to "Functional groups in fiber samples were determined by'
4. Authors should explain reason(s) for the difference observed in FTIR analysis of PLA modified with ADO.
5. I doubt if there exist any functional group like -COO as claimed by the authors in line 384. The functional groups that can be linked with carboxylate are -C=O and -C-O.
6.Figure 16 is missing.
Comments on the Quality of English LanguageThe English language is fine, requires only minor modification.
Author Response
Hello,
thanks for reviewing my manuscript. All the comments and suggestions were very helpful and I began to use them. Thank you for your work and your time to read and improve my research work.
The first standard refers to the environmental conditions of mechanical analysis and the second to the analysis of specific parameters. Corrected:
“Mechanical properties of fibers were determined under standard environmental conditions of 20 ± 2 °C and RH = 65±4%, according to the PN-EN ISO 139:2006 standard. The mechanical parameters such as linear density, breaking tenacity, and elongation at break were estimated using the PN-EN ISO 2062:2010 method A on an Instron 5544 Tensile Tester (Norwood, MA, USA).”
The last sentence in the FTIR-ATR paragraph is an explanation:
“This band for fibers modified with ADO is more complex, which may indicate an interaction of the carbonyl group of polymer with the plasticizer. This region (1700-1800cm-1) is sensitive to the conformation at the chain and it can be used to differentiate the crystalline form or crystals of PLA.” But added also: The change in the region 3100-2600 cm-1 for ADO fibres may indicate an interaction of the modifier with the methyl group in PLA.
Corrected: The publications indicated that, at approx. 1600 cm-1, a band appears after degradation and is assigned to the carboxyl ion (-COO-), formed as a result of the hydrolysis of the ester group [64-66].
Inserted graphs that disappeared from the manuscript. All corrections made in accordance with the reviewers' suggestions are highlighted in red.
Best regards
Karolina Gzyra-Jagieła
Reviewer 3 Report
Comments and Suggestions for Authors
Manuscript Title : Influence of low molecular esters on melt- spinning and structure of poly(lactic acid) fibres
Manuscript ID : materials-2674205.
The present study investigated the development of melt spun modified PLA fibers. Then, modified fibers were analyzed by various techniques (DSC, FTIR, WAXD and GPC/SEC) to assess their structural and thermal properties. In addition, mechanical tests and evaluation of the morphology of fibers using SEM microscopy were performed.
The topic discussed in this paper is very interesting and the experimental work is important. Results found in this research work are of great interest. Nevertheless, the language of the manuscript is poor which reduced the paper quality.
The paper could be considered for publication in Materials journal only after authors address the below given issues:
* Explain in a clear manner the novelty of the present work.
* Improve the English of the manuscript, particularly in the introduction section.
* A lot of literature references are very old; some of them could be replaced by more recent ones.
* Detailed comments are reported on the attached file : Review-materials-2674205.pdf

Comments on the Quality of English LanguageEnglish needs extensive editing by an expert.
Author Response
Hello,
thanks for reviewing my manuscript. All the comments and suggestions were very helpful and I began to use them. Thank you for your work and your time to read and improve my research work.
All suggestions from the pdf file have been implemented.
Article revised with reviewer's guidelines, additionally checked by English teacher but the language style has been improved againe.
Added:
''Polylactic acid (PLA) is a versatile material commonly used in medicine. It can be produced using two methods: melt-spun technology, which is similar to the production of other melt-spun fibers like polypropylene, or solution spinning (both dry and wet) [19]. PLA is not only used for absorbable sutures, but its fiber braid can also be used in human tissue repair and as a drug delivery system. During the material's degradation process, pharmaceuticals can be released successively [20-21]. Additionally, the fiber can be used to create structures such as surgical meshes used in bone tissue engineering or as an Achilles tendon [22-25]. For the spinning process, the crystallization and degradation processes are important aspects, as they influence the mechanical properties. These aspects, however, determine the applicability of the material. Therefore, understanding the spinning process is crucial. In the medical field, knowledge of the mechanisms allows the development of a fibre material for an application. It is particularly important to be able to design medical fibres so that the modifier shows a dependence on a key parameter, e.g. strength, degradation time, elasticity, etc. The fibres for suture threads for dermal, hypodermic or muscular applications will require completely different requirements. Therefore, research into modifications and their effects on the properties of the material from a medical perspective is important.''
* A lot of literature references are very old; some of them could be replaced by more recent ones. -The new publications refer to the old ones and they are the source of this knowledge.
Inserted graphs that disappeared from the manuscript. All corrections made in accordance with the reviewers' suggestions are highlighted in red.
Best regards
Karolina Gzyra-Jagieła
Round 2
Reviewer 2 Report
Comments and Suggestions for Authors
Accept
Reviewer 3 Report
Comments and Suggestions for Authors
The authors kindly did a great effort revising the present paper. They addressed the issues that I requested. In addition, they added useful paragraphs and literature references, so the objective of the research work is now clear and the manuscript well structured. Moreover, the spelling mistakes of the manuscript were checked and corrected, so as this version is well written in an easier language for readers.
I suggest that the paper could be published in Materials journal in its present form.